# Navigating Infodemics, Unlocking Social Capital and Maintaining Food Security during the COVID-19 First Wave in the UK: Older Adults’ Experiences

**DOI:** 10.3390/ijerph18147220

**Published:** 2021-07-06

**Authors:** Heather Brown, Kate Reid

**Affiliations:** School of Psychology, University of Glasgow, Glasgow G12 8QB, UK; Kate.Reid@glasgow.ac.uk

**Keywords:** older adult, COVID-19, food insecurity, loneliness, social connectedness

## Abstract

In March 2020, a national UK lockdown was implemented in response to rapidly rising COVID-19 infections. Those experiencing the most severe public health restrictions were ‘shielding’ groups as well as those over 70 years of age. Older age adults, many of whom were active, independent, and socially connected were immediately instructed to stay at home, to limit all external social contact and consider contingency for maintaining personal food security and social contact. The purpose of this qualitative study was to explore the experiences of older adults during the first UK lockdown (March–June 2020), specifically how our sample reacted to public health messaging, staying food secure and drawing on available social capital within their community. Semi-structured telephone interviews were conducted with eight participants. In addition, twenty-five participants completed a qualitative ‘open-ended’ survey. The data was collated and analysed, adopting a Thematic Analysis informed approach. Three themes were identified: (1) *Too Much Information*, (2) *The Importance of Neighbours and Connections* and (3) *Not Wishing to be a Burden*. These findings offer a rich insight into how early lockdown measures, never witnessed since World War 2, exposed existing pre-pandemic inequalities and concerns relating to loneliness, isolation and wellbeing. The findings are of relevance to researchers, older adult advocate groups and policy makers to inform post COVID-recovery within communities to ensure healthy ageing.

## 1. Introduction

The UK population is ageing with people aged 75+ projected to be the fastest growing group in Scotland [1]. The Scottish Government policy document, Reshaping Care for Older People: A Programme for Change 2011–2021 [2], sets out the vision that ‘*older people are valued as an asset, their voices are heard, and they are supported to enjoy full and positive lives in their own home or in a homely setting*’.

Staying food secure and participating in meaningful social connections are basic, yet fundamental indicators of leading full and positive lives and directly adhere to principles of healthy ageing [3]. On the 16 March 2020, in response to accelerating rates of COVID-19 infections and hospital admissions, government led public health advice recommended that those over seventy should stay at home, avoid leaving the house and avoid close physical social contact with friends and family who were not living with them [4]. At this point, it became an urgent national and local government priority to try and mitigate the potentially negative mental and physical health consequences resulting from this need for rapid, unprepared withdrawal of older adults/shielding groups from their formal and informal support structures [5]. The following week, on the 23rd of March, a national ‘lockdown’ was implemented across the UK. The psychological health implications facing older adults were unknown at this time, but it was probable, given previous research [6] that isolation and loneliness may disproportionately affect this older age group.

### 1.1. Food Access and Security

Older adults face heightened risk of inequalities which can be evidenced in terms of food insecurity and malnutrition [7]. Food Insecurity is defined as a lack of reliable access to enough safe and nutritious food to support normal development, functioning and activity [8]. However, identifying and operationalising household food insecurity, benefits from a multi-dimensional approach, where the inability or uncertainty to acquire and consume an adequate quality and quantity of food in socially acceptable ways is required [9]. The British Dietetic Association in July 2020 organised a letter on behalf of its members to the Secretary of State for Health and Social Care stating that the ongoing social distancing and shielding measures would continue to restrict normal access to food and therefore lead previously food secure older adults towards a “perfect storm” of malnutrition due to COVID-19 public health measures [10].

A number of policy interventions were created for supporting access to food and preventing food insecurity such as food box schemes and priority online food shopping which were created across the UK [11]. However, there was concern raised towards many of these schemes, especially around delays and the nutritional quality of the food [12] and the need for digital literacy. Many of these targeted initiatives ended in late July 2020 meaning those who wished to continue to shield during the ‘second wave’ in the autumn of 2020 may have had challenges in accessing food [11]. Charities and social enterprises such as The Food Train have stepped up to offer food delivery and meal making services for older adults, often making the most difference within communities of older age adults, where government support was slow or inadequate to identify and respond to existing and newly exposed social care needs, including food security [13].

### 1.2. Loneliness and Social Connectedness

Along with the need for secure access to food, social relationships are a crucial psychological need, integral to well-being and healthy ageing. The connection between socialisation and health has been demonstrated for older adults where unwanted social isolation and loneliness are associated with an increased risk of many health conditions including increased blood pressure, reduced immunity, depression, impaired cognitive function, sleep issues and mortality [14].

Proximal neighbours dwelling immediately next to or near the homes of older adults are a resource often overlooked within research where social connections have been evidenced through indicators of quality and quantity of family members and close friends [15]. However, if physical bonds between family members or friends are disrupted or indeed severed, then neighbours can be an important source of support and access, to what is known as ‘social capital’ [16]. According to Seifert & König [17], if older adults have ties to their neighbours then this can reduce their sense of isolation and help individuals link to external social capital. This social capital can come in many forms such as equipping with knowledge of and even linking to local social networks and community assets including local volunteer groups and social enterprise/charities operating in the area. There has been an increased interest in the association between neighbourhood connections and the health and wellbeing of older adults. Social gerontology research [18], has highlighted the importance of neighbourhood cohesion and feelings of belonging on the quality of ageing in communities. Elliott et al. [19], reported a moderate association between the perceptions of neighbourhood cohesion and mental wellbeing within a cohort of older adults. This also extends to issues of Food Security where an understanding of the ‘material conditions’ an older adults faces, such as financial resources, quality of family and social connections and the realities of ageing (increased illness and disability) can significantly impact on perceived food security in the home [13].

With the arrival of a pandemic, it was important that older adults retained and adapted to new ways of connecting with available social connections as well as accessing the new and necessary social capital to enable healthy ageing in place to continue [20]. The psychological and physical cost of public health measures such as social distancing, travel restrictions, non-essential shop closures and disbanding of local lunch-clubs and befriending activities offer an important backdrop to understand how older adults have navigated this unprecedented time.

### 1.3. Rationale

The current research study aims to explore and document the experiences of those aged 70 and over in the UK on maintaining food security and feeling connected during phase 1 (lockdown) and the beginning of phase 2 (initial early public health easing from lockdown). Qualitative research was well suited to researching this novel area of study as it enabled opinions, experiences and feelings of the participants to be investigated whilst giving a unique depth of understanding to the field of enquiry. The study sought to answer the following questions:How are social distancing and shielding practices impacting on the perceptions of food insecurity for those seventy and over?How are older adults maintaining social connectedness while remaining shielded from family and friends?What immediate ways can we, as a society, improve the lives of older adults, while ensuring adherence to social distancing at this time of COVID-19?

## 2. Materials and Methods

### 2.1. Sampling and Participants

Thirty-three (see Table 1), older age (70+), community dwelling adults were recruited. Eight participants consented to take part in one-to-one telephone interviews, seventeen consented to fill in an open-ended survey sent in the post and eight consented to fill in an online form. The inclusion criteria for the study were that all participants had to be 70 or older, living in the community and not suffering from any cognitive impairments such as dementia. Participants were recruited via several different methods. Some participants were recruited online via community social networks such as ‘self-isolation’ Facebook groups, which had emerged as a result of the pandemic. Participants were also recruited through a GP surgery where a phone text was circulated to all patients aged 70 and over. The rest of the sample were recruited via convenience sampling.

### 2.2. Data Collection

After indicating willingness to be involved in the study the researcher contacted participants via telephone. The prospective participants were then asked whether they wished to take part in a telephone interview or fill out a survey (which contained representative questions used to support the one-to-one interviews) in the post or online. Maximum flexibility in delivery and retrieval of qualitative data was considered essential as the study was taking place during the early months of the pandemic and we did not wish to assume that all participants would be willing to be contacted via telephone or complete an online survey. One-to-one semi-structured interviews, lasting up to 30 min were recorded on a secure electronic device with data stored in a GDPR compliant method. The qualitative data was transcribed verbatim by the researcher.

Participants who wished to fill out the hand-written postal survey, with ‘free text’ open ended answers, were sent out the survey and when completed they returned it to the researcher in a pre-paid envelope. Participants who completed the form online (via GDPR compliant survey software) did so independently and were provided with a link to a Microsoft Form.

The study questions were designed to explore how participants had experienced lockdown, specifically looking at access to food, staying socially connected and how they were coping. The questions were theoretically informed and constructed, based on review of current research.

### 2.3. Ethics

The study received ethical approval from the College of Science and Engineering Ethics panel at the University of Glasgow. All participants were provided with a participant information sheet and informed consent (verbal or written) was obtained prior to taking part. During the one-to-one interviews, consent was approached as a process and as such the participants were given opportunities to reaffirm consent during and at the end of the interview. Pseudonyms have been used and any other identifiers, for example names of family and places have been omitted or changed. All the participants were left with a full list of organisations and services available to them at this time if they were experiencing hardship and/or had questions about the pandemic. The researcher only exited the interviews if they were satisfied the older adults were not facing acute crises.

A detailed reflexive diary was kept by the researcher during the data collection and analysis, enabling initial coding as well as accommodating thoughts and ideas concerning the data to be noted. The reflexive note taking also helped refine and orientate the research questions as some were not as fully explored from the perspective of participants. It also enabled the researcher to be mindful of potentially sensitive or upsetting disclosures given by the participants.

### 2.4. Analytical Approach

Thematic Analysis is a method for identifying, analysing and reporting patterns known as themes within the data [21]. The data was analysed following Braun and Clark’s [21], 6-phase guidelines. During phase 1, the data from all 33 participants (both those who took part in the interviews and those who filled in the survey) was transcribed. The transcripts were read through a number of times and any initial ideas (potential codes) noted. In phase 2 the verbatim data was worked through systematically and any interesting features of the data were identified and organised into meaningful conceptual clusters of codes. For example, participants mentioned the media coverage of the pandemic. Therefore, the initial codes of ‘overwhelming’, ‘watch less’, ‘anxiety’, ‘distress’ and ‘excessive, alarming coverage’ were created. Phase 3 started with a long list of codes which were collated and sorted into potential themes. For example, the potential themes of excessive information, reducing exposure to information and annoyance at the coverage were created (see Appendix A Table A1). In phase 4, the potential themes were refined and candidate themes for analysis were identified. For example, the potential themes related to information and public health advice were grouped together to form the main theme of ‘Too Much Information’. In phase 5, the themes were further refined based on the specifics of each theme and how they link to the overall story of the analysis including alignment with the research questions. Attention was paid to documenting both semantic and latent level themes to offer both description and conceptual interpretations based on the rich data from the participants. The final phase, phase 6, involved selecting example quotes from participants which evidenced the themes and which demonstrated close alignment to inform the research questions. An example of the process behind developing the final themes can be seen in Appendix A.

## 3. Results

The following three themes were identified through Thematic Analysis as consequences of the restrictions requested of older adults (see Table 2). The analysis is presented with the aid of thematic headings, supported with verbatim evidence from the transcripts. The analyst’s narrative is used to anchor the themes and evidence into an account that highlights a range of lived experiences which are central to the research questions and topic area. Pseudonyms are provided.

### 3.1. Too Much Information

When participants were asked about their initial, early exposure to the fast-paced unfolding COVID-19 pandemic, the common experiences were situated around the issue of negotiating and regulating the continuous news and communications.

“*I think there’s been too much TV and airtime given to reporting, dissecting and discussing the virus. Total overload and spreading fear unnecessarily.*”[Mildred 88–89]

In this extract, Mildred stated that she felt the communication from the media was excessive and could be improved. The use of the words “dissecting” and “discussing” suggested her perception was that of uncertainty and debate around the limitless amount of information aimed at the public at this acute time. Mildred’s use of the word “overload” suggested that she was overwhelmed by this breadth and depth of coverage, which in her view seemed to be promoting ‘unnecessary’ fear. Issues of trustworthiness surrounding the sources of communication indicates that even at an early point in the pandemic, some individuals were not equipped or secure in their own perceptions of who was a credible source. This is seen in a comment by Gary.

“*I think it tends to make me anxious…Well you know, you begin to wonder if you can trust anybody.*” [Gary: 150–152]

Mechanisms to self-regulate news and communications about the pandemic included ‘tuning out’, where many participants were choosing to limit their exposure to the media. One example of this came from Lola.

“*A lot of my friends, we just look at the headlines, listen to the headlines, we don’t want to hear anymore.*” [Lola: 213–215]

Lola suggested that this was a shared feeling and behaviour among her friends. Avoidance as a coping mechanism offered an effective way to limit exposure to news and communications, often motivated through a belief that mental health was becoming compromised during this early point in the pandemic.

Whilst some of the participants started to lean away from traditional sources of dissemination such as television and online media, it opened up space to explore more informal networks of information and support located within existing or newly formed community networking such as a renewed role for the local neighbour.

### 3.2. The Importance of Neighbours and Connections

When investigating how older adults had stayed socially connected during the COVID-19 pandemic, it was anticipated that they would mainly identify family and existing close friends. Participants did reinforce the importance of established familial and social networks but also introduced the increased availability of connecting or re-connecting with neighbours and people in the local community, often previously unknown to them. Twenty four (72%) participants mentioned talking with their neighbours as a way of staying socially connected. The COVID-19 pandemic and the steps taken to find help and support highlighted the importance of social connections and community. This was highlighted in a comment by Edith.

“*One of our neighbours way at the beginning before the lockdown came round and put a note through everybody’s door with their phone number and asked about setting up a WhatsApp group and would we be interested? … it’s quite handy for any kind of local news … I just think it is really kind.*”[Edith: 261–268]

Edith’s comment indicates that she really appreciated the proactive gesture of her neighbour and that she felt more secure because of it. The disproportionate impact on older adults, including those living alone, separated from family at this time had meant that acute concern was raised in the local community, with help facilitated by leaflet dropping, online social networking and mobilisation of existing social care charities.

“*People are on their balconies. I’ve managed to speak to them, I speak to anybody so with doing that, it’s really helped me.*”[Mary: 202–204]

Mary speaks of neighbours who were more accessible and had time to chat. Her account indicates the importance of spontaneous conversations and she enjoyed the novelty of this. She stated that speaking to other people helped her cope with the difficult situation. Similarly, Mildred reinforces the need for social connections, especially those brought about by living in supported accommodation.

“*Where I stay is an apartment complex for retired active people. So as we come and go there is always someone to say “hello” to–which alleviates the thought of isolation.*”[Mildred: 53–55]

Mildred’s account offers a sense of cohesion and community; the use of the word “we” suggests she felt a part of her immediate community. The company of neighbours who are also her peers would have added to this sense of belonging.

“*My neighbour upstairs who comes when she goes out for a walk, she’s in her seventies, she goes for a walk every day and she always comes to the window and we have a chat.*”[Lola: 162–164]

Again, Lola’s comment shows the importance of having neighbours that are also your peers and it is clear that they were making an effort to chat to her and check-in. During this time the ways in which neighbours could communicate were limited and talking through windows, over garden walls and across hedges was a very important way of having proximate, meaningful contact with someone whilst remaining safe.

The participants reported that neighbours had been very kind, supportive and helpful, however, it did not stop some from worrying that they may be a burden. This was especially the case in terms of any transaction taking place, such as accepting help with food shopping.

### 3.3. Not Wishing to Be a Burden

Several participants commented on not wishing to be a burden to others when it came to food access. All who commented on this mentioned that they would compromise on the food offered to them or brought to them by neighbours.

“*I feel a bit uncomfortable as they won’t take payment for the shopping so I compromise on what I really want.*”[Harriet: 29–30]

Harriet was uncomfortable being perceived as someone who required charity and basic support to access food. Asking for help to improve access to food can change the power dynamics within normal relationships and for Harriet this might have taken away a sense of pride and control. It is also worth noting that Harriet’s sense of food insecurity was not based on financial pressure but more simply, the inability to go out to shop due to shielding public health advice. Harriet’s comment illustrates the unintentional consequence of an act of kindness, which potentially had a negative impact on the quality and quantity of the food she requested. Similarly, Joan states:

“*I suppose my meals are a bit boring. I try to keep my shopping list simple, so that they can be done without looking around for unusual items etc. i.e., I don’t want to waste their time.*”[Joan: 56–58]

Joan did not wish to waste the time of her neighbours who helped buy her food shopping. The “unusual items” she commented on may be treats or the preferences, particularly to her own tastes, that she might buy for herself. However, her comment suggested she was willing to restrict these personal preferences for the sake of simplicity and to relieve the pressure of feeling a burden. This resulted in her meals being less interesting and enjoyable.

“*I can’t help but feel beholden-but this is presumably one of the penalties of being old!! i.e., to accept gracefully!*”[Joan: 46–47]

A further comment by Joan suggests that she felt that there must be some negative consequence of growing older. The striking use of the word “penalty” indicates a belief that there has to be some trade-off to getting to live to an older age. Joan may associate ageing with a loss of independence and a dependence on others. As stated earlier being food secure requires being able to access food in a socially acceptable way [22]. Some of the participants stated that they worried that they may be a burden on others, especially individuals who are not bound by family ties, and this did seem to be impacting on their food choice. They were willing to compromise not just on the quality of the food but also the quantity. Therefore, they were not food secure.

## 4. Discussion

The aim of this research was to explore how older adults maintained food security and social connectedness during the first lockdown of the COVID-19 pandemic. The older adults in this study gave unique perspectives to the current situation with three themes being identified (see Table 2). These informed the research questions which aimed to investigate how social distancing and shielding impacted food insecurity, how older adults were maintaining social connectedness and how society can improve the lives of older adults at this time. The findings of this study lend support to those of previous research, highlighting the wide variety of experiences in relation to healthy ageing and offer novel perspectives specifically during the early months of the COVID-19 pandemic [23].

During any crisis, but specifically a novel health crisis, the public are reliant on the local and national government, public health officials and the media to convey accurate public health information [23]. This enables people to try and understand the situation and to align their own decision making in accordance with public health advice. During this pandemic, rapid and complex public health information, likely to cause concern and alarm was present across all media and social networking outlets. When investigating how participants felt society could improve their lives during this time the idea that the information was excessive was discussed as evidenced by the theme “*Too Much Information*”. This information overload has been termed by Fiorillo and Gorwood [24] as an “infodemic”.

How information is managed and released formed a large part of the experiences as told to us through interviews. For some in our sample, there was a belief that the volume of public health advice was excessive and alarming. According to Garfin et al. [23], the public should be advised to avoid speculative stories and to reduce their exposure to the repetitive media stories. In the current research, participants appeared to be negotiating their exposure by their own volition by using avoidance coping techniques in order to protect their well-being while also trying to be aware of what public health measures were required. Avoidance Coping involves denying, minimising and avoiding dealing directly with the stressor and is a mechanism of coping that allows an individual to deal with sources of stress [25]. Although this coping mechanism is linked to distress and depression [26], it has been shown to be effective in reducing acute stress in certain situations specifically those that are novel and out-with the control of the individual [25]. Hence, it may have been a helpful ‘holding’ strategy for participants during this time. Similar results have been found during previous pandemics, for example in a qualitative study investigating stigmatisation and coping during the H1N1 pandemic [27]. Information overload and the use of avoidance coping were therefore not unique to this pandemic.

The theme “*The Importance of Neighbours and Connections*” illustrated the way that older adults were staying socially connected while experiencing social distancing measures. In this new environment, often devoid of regular physical visits by external family and friends, neighbours became a source connecting older adults to company and in some cases, access to local news and support in the form of shopping– social capital. Close to three-quarters of participants mentioned neighbours when asked how they stayed socially connected. This suggests that neighbours were important to them during this time and helped them remain connected while ‘shielding’. This is in line with research from previous public health emergencies which has shown that neighbours often play a critical role in responding to the needs of those who are vulnerable [28] and in the response, resilience and recovery of a community following any crisis [29]. In this study virtual contact with family remained important, but neighbours had the potential to play a leading role in the physical lives of the older person where they were often one of the few people participants were able to have non-virtual contact with. Findings in this study also confirm previous qualitative research [30], which found that older adults stressed the importance of just knowing that support was close by even if they did not need to make contact. This is an example of community cohesion.

The nature of the crisis highlighted the importance of community cohesion in enabling vulnerable older adults to harness the links to social capital in the form of staying connected to others while being asked to ‘shield’. The findings from this study support existing research which has shown that having ties with your neighbours can help reduce isolation and help older adults cope with everyday life which became increasingly stressful and difficult during lockdown [17]. Our research adds to the evidence base where the role of the neighbour can be dynamic, adaptive and critical to offering ‘first response’ to local older adults–and no doubt, other age groups vulnerable to inequalities during times of need.

Perceptions of age and ageism were evidenced in this pandemic with concerns raised over the portrayal of high rates of death in older age adults, often in care home as somehow inevitable, predictable and acceptable within a pandemic [20,31]. All of this devalues older adults and is likely to contribute to feelings of worthlessness [20]. This is evidenced by the themes “*The Importance of Neighbours and Connections*” and “*Not Wishing to be a Burden*”. This reflects the results of previous studies which have shown that neighbourhood cohesion can improve feelings of belongings and improve quality of ageing [18]. There is a growing body of research which demonstrates strong links between neighbourhood characteristics and mental wellbeing [32,33,34], suggesting that having a positive sense of ‘neighbourhood’ can help provide a positive sense of identity for older adults. This might help explain the link between neighbourhood cohesion and wellbeing [19]. However, participants in this study were still worried that they were a burden to those helping them.

There has been an increase in food insecurity attributable to the pandemic where older adults previously food secure have experienced acute food insecurity, particularly during the first lockdown in 2020 [35]. For example, the pandemic has made it more difficult for those 70 and over to access food as they were asked to shield and have food delivered to their home [36]. In this study, we were able to document how older adults in our sample were at times hesitant to ask for help or when help was asked for in the context of food access, compromises were made to protect their concern of being seen as a burden or asking for ‘too much help’. Although all the participants in this study managed to access food, just over 1/3 decided to leave their house to get their food shopping themselves, despite it being against the advice given by the government. Therefore, by “Not Wishing to be a Burden” some participants chose to put themselves at a potentially higher risk of contracting COVID-19 by going to get their food shopping themselves rather than asking for help or risking becoming food insecure. We know from previous research that asking for food can change the power dynamics within relationships especially when attempting to protect dignity [7]. Exchanges that are unequal, such as asking for help with daily activities can lead to distress and increase the risk of depression [37]. Farrer [38], highlighted that retaining independence over decision making about food access and food choice is one of the most important activities older adults wish to retain. Therefore, feeling a burden while carrying out this activity may have impacted negatively on participants well-being. Supported food services such as The Food Train offer a good model for understanding how to maintain dignity in supporting older adults to be food secure at home [39]. Food insecurity was clearly an issue for some participants in our study as demonstrated by participants compromising on their food choice.

This research was conducted at an early point in the UK COVID-19 pandemic. Recruitment was carried out by several different methods; through social media, word of mouth and through a local GP surgery. As a result, it was potentially harder to recruit an ethnically and socially diverse sample. Interviews had to be carried out over the phone or via an open-ended survey. There are a number of advantages to this, e.g., allowing participants to carry out the interview in their own home making them feel more in control and empowered [39] and that the research was not limited by geography, therefore people with physical difficulties and mobility problems could easily take part. However, relying on digital literacy (using a computer) combined with the need for good communication skills to support being interviewed by the researcher on the phone did pose challenges to reach those most marginalised and isolated within this population. Future research could compliment this study by offering an insight from the perspective of those who stepped-up in their local community to support the needs of their older neighbours, to better understand what motivated them, how they have remained engaged in local community activism and how we build back more socially and community focused intergenerational neighborhoods.

## 5. Conclusions

The aim of this research was to investigate older adults’ experiences during COVID-19 specifically looking at staying food secure and socially connected. In line with previous research the importance of neighbours and community cohesion was instrumental in supporting those most in need and most at risk of facing inequality. How we continue to harness the good will, acts of kindness and community cohesion as we build back from COVID-19 are larger societal questions that remain to be addressed. We can see that over the space of a few short but life changing months, communities can mobilise and help those most in need supported by social media, local knowledge, quality health and social care and good will.

## Figures and Tables

**Table 1 ijerph-18-07220-t001:** Summary of Demographic Data for Participants.

Demographic Information		N
Gender	Female	27
	Male	6
Ethnicity	White	33
Living Situation	Alone	22
	With Partner/Spouse	10
	With Child/Children	1
Access to Outside Space	Yes	32
	No	1
	Very Good	6
	Good	15
Health in General	Fair	11
	Bad	1
	Very Bad	0
	Family/Friends	18
Method for Accessing Food	Home Delivery Service	7
	Left house to get their food shopping themselves	13
Total		33

**Table 2 ijerph-18-07220-t002:** Summary of Themes.

Theme Title	Theme Definition
1. Too Much Information	Discusses the impression that the information being conveyed from government and media sources was “too much” leading participants to feel anxious, annoyed or disengaged. In response to this many decided to reduce their exposure.
2. The Importance of Neighbours and Connections	Discusses the increased contact participants had with neighbours and how important this was in staying socially connected.
3. Not Wishing to be a Burden	Discusses the theme that some participants felt a burden to those helping them access food. Therefore, they would compromise on the quantity and quality of the food they requested.

## Data Availability

The data that support the findings of this study are available from Brown, H., & Reid, K. (5 July 2021) Transcripts. Retrieved from http://osf.io/6jpvn.

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
