# Peer review of "Navigating Infodemics, Unlocking Social Capital and Maintaining Food Security during the COVID-19 First Wave in the UK: Older Adults’ Experiences"

_ijerph, 2021, doi:10.3390/ijerph18147220_

Round 1

Reviewer 1 Report

This research has an original objective and a meaningful content. I congratulate the authors for the work done. I am grateful with the editors for the possibility of revising this manuscript. Although the quality of the manuscript is high, I would like to make some contributions that I hope will increase it and improve readers' understanding.

Introduction

The introduction is clear and well worked.

Materials and Methods

Study design

The study design is appropriate and well described.

  1. Materials and Methods 2.1. Sampling and Participants

Does the recruitment of study subjects comment that it was carried out by several different methods, social networks? For a study of this type, I believe that patients should be better selected based on certain inclusion or exclusion criteria that are not defined by what the sample of 33 participants was via social networks, it seems to me too scarce lines 117-125.

Data analysis

Statistical analysis is correct and well described.

Discussion

Discussion is well oriented.

Author Response

In order to address the comments an additional statement has been added under the sampling and participants section in the methods which states the inclusion criteria for the study as this was not previously mentioned. The description of how participants were recruited (123-128) has been edited to make it clearer for readers. An additional sentence has been added in the discussion (440) to reiterate the different methods used for recruitment.

Reviewer 2 Report

The paper is laid out clearly in a way that helps the reader understand how the study was conducted. The article would be strengthened by drawing stronger links about food security/insecurity to the ways in which older adults experienced this during COVID sheltering.  For example, I was left asking how (other than receiving food from family and friends) was food security/insecurity experienced?  Did the adults receive food from families and friends and fix their own meals, or receive meals already prepared, and were there mobility needs other than the circumstances of COVID keeping participants away from the grocery stores- since approximately one third of the study participants went to the stores themselves to buy food.  It would be interesting to know more about the relationships of the adults to whomever was supplying the food to help the reader understand why the lists were kept simple or short, and who was providing food free of charge?  In spite of the order to shelter in place, relationships between adults and the friends/families supplying the food would be interesting to know about because if relations were strained before the pandemic, they would be strained during the pandemic, particularly if more was being asked of family/friends in terms of supplying food.  Although the central theme of the paper did not hinge on relationships, a further explanation of the food exchange would be helpful to justify the discussion.

Author Response

In order to address the comments from the reviewer edits have been made to increase the strength of the link between participants and food security/insecurity. In the discussion (422-427) we have added an explanation around the fact that around 1/3 of participants chose to leave their house in order to access food themselves discussing how participants chose to leave their house against government advice to make sure they had access to food or because they did not wish to ask for help. An additional sentence has been added around local food services and maintaining dignity when supporting older adults to be food secure.

Unfortunately we did not collect the data which would allow us to investigate the relationship between participants and those who supplied their food or the relationships between participants and their families. Therefore we are unable to address this comment although it would be a very interesting area to investigate.

Round 2

Reviewer 1 Report

modifications made by the authors are accepted